# Ubiquinol Supplementation Alters Exercise Induced Fatigue by Increasing Lipid Utilization in Mice

**DOI:** 10.3390/nu11112550

**Published:** 2019-10-23

**Authors:** Huan-Chieh Chen, Chi-Chang Huang, Tien-Jen Lin, Mei-Chich Hsu, Yi-Ju Hsu

**Affiliations:** 1Graduate Institute of Sports Science, National Taiwan Sport University, Taoyuan 33301, Taiwan; 1051302@ntsu.edu.tw (H.-C.C.); john5523@ntsu.edu.tw (C.-C.H.); 2Department of Neurosurgery, Taipei Medical University-Wan Fang Hospital, Taipei 11696, Taiwan; trlin1@hotmail.com; 3Taipei Neuroscience Institute, Taipei Medical University, New Taipei City 23561, Taiwan; 4Graduate Institute of Injury Prevention and Control, College of Public Health and Nutrition, Taipei Medical University, Taipei 11031, Taiwan; 5Department of Sports Medicine, College of Medicine, Kaohsiung Medical University, Kaohsiung 80708, Taiwan; 6Kaohsiung Medical University Hospital, Kaohsiung 80708, Taiwan

**Keywords:** Q10, exercise performance, anti-fatigue, glycogen storage

## Abstract

Ubiquinol (QH), a reduced form of coenzyme Q10, is a lipid antioxidant that is hydro-soluble and is commonly formulated in commercial supplements. Ubiquinol has been increasingly reported to exert antioxidant functions, in addition to its role in the cell energy-producing system of mitochondria and adenosine triphosphate (ATP) production. The aim of this study was to assess the potential beneficial effects of QH on anti-fatigue and ergogenic functions following physiological challenge. Forty 8-week-old male Institute of Cancer Research (ICR) mice were divided into four groups (*n* = 10 for each group): Group 1 (vehicle control or oil only); Group 2 (1X QH dose or 102.5 mg/kg); Group 3 (2X QH dose or 205 mg/kg); Group 4 (6X QH dose or 615 mg/kg). Anti-fatigue activity and exercise performance were studied using the forelimb grip strength experiment and exhaustive weight-loaded swimming time, and levels of serum lactate, ammonia, glucose, BUN (blood urea nitrogen), creatine kinase (CK), and free fatty acids (FFA) after an acute exercise challenge. The forelimb grip strength and exhaustive weight-loaded swimming time of the QH-6X group were significantly higher than those of the other groups. QH supplementation dose-dependently reduced serum lactate, ammonia, and CK levels and increased the FFA concentration after acute exercise. In addition, QH increased the liver and muscle glycogen content, an important energy source during exercise. Therefore, the results suggest that QH formulation is a safe dietary supplement for amelioration of fatigue and for promoting exercise performance.

## 1. Introduction

Physical fatigue is commonly defined as the reversible decline of performance during activity [1]. The commonly known fatigue mechanisms are related to metabolic fuel availability and the accumulation of waste products. Reactive oxygen species (ROS) are produced by practicing intense and long-term physical exercise, which may induce tissue damage and oxidative stress [2], and antioxidant supplementation attenuates exercise-induced oxidative stress and fatigue of the body [3]. It is reasonable to assume that if one can overcome the effects of fatigue, its impact upon physical performance can be reduced. Coenzyme Q10 (Ubiquinone, CoQ10), a lipid-soluble vitamin-like nutrient found naturally, is synthesized as an endogenously antioxidant in the body [4]. Due to its ubiquitous distribution in nature, CoQ10 is also known as ubiquinone [5]. In mitochondria, CoQ10 is a key component of the mitochondrial respiratory chain for adenosine triphosphate synthesis [6], and it is found in both the reduced and oxidized states [7]. The two main functions of CoQ10 are energy production and antioxidation. CoQ10 participates in aerobic cellular respiration and is a cofactor in the mitochondrial electron transport chain; therefore, it plays an essential role in the production of adenosine triphosphate (ATP) [8]. CoQ10 is mainly biosynthesized and concentrated in tissues and organs with high metabolic turnover or energy demand, such as the heart, kidneys, liver, and muscles [9]. CoQ10 also shares its biological implication with membrane-associated related redox, which serve as other redox carriers, including dehydrogenases, cytochromes, and non-heme-iron proteins [10,11]. Increasing attention has been paid to the application of CoQ10 as an antioxidant for the prevention and treatment of a great variety of functional disorders, such as fatigue.

CoQ10 has two major disadvantages. First, CoQ10 is a relatively large hydrophobic molecule; its absorption is often slow and limited [12]. CoQ 10 mainly occurs in the small intestine, passes into the lymphatic system, and finally to the blood and tissues [13]. Second, ubiquinol is the reduced active antioxidant form of Coenzyme Q10. Although the oxidized form of ubiquinone is normally readily reduced to ubiquinol enzymatically after dietary intake [14], research has shown that this process becomes increasingly difficult with age [15,16]. 

Some studies have investigated the potential ergogenic value of CoQ10, but the results have been inconsistent. The differences between the results may be due to the distinct study designs, which differed in terms of type, dosage, and time frame of treatment. The reduced and more absorbable form of CoQ10, ubiquinol (QH), and the possible use of CoQ10 as a natural nutritional supplement for exercise performance and to mitigate fatigue have not been examined. In this study, we investigated the effects of QH supplementation on exercise performance and on anti-fatigue activities in a mouse model.

## 2. Materials and Methods

### 2.1. Materials, Animals, and Treatment Design

QH was purchased from Sinphar Pharmaceutical Co. Ltd (Yilan, Taiwan). Forty 8-week-old specific pathogen-free (SPF) male Institute of Cancer Research (ICR) mice were used in this study. All the experimental animals were given free access to a commercial standard chow diet (No. 5001, PMI Nutrition International, Brentwood, MO) and distilled water ad libitum. They were maintained under standard laboratory conditions of 12-h light and dark phases at a 22 ± 2 °C ambient temperature with 50%–60% relative humidity. Twice each week, the bedding and bottles were changed and cleaned. All the experiments were conducted under a protocol approved by the Institutional Animal Care and Use Committee (IACUC) ethics committee of the National Taiwan Sport University, and the study conformed to the guidelines of protocol IACUC-10516.

The detailed experimental procedure is illustrated in Figure 1. After a 1-week acclimation period, mice were randomly distributed into the four groups (*n* = 10 per group) for oral gavage treatment with QH or oral gavage of vehicle only once per day for 28 days. The groups received the following treatments: Group 1 (vehicle control or oil only); Group 2 (1X QH dose or 102.5 mg/kg); Group 3 (2X QH dose or 205 mg/kg); Group 4 (6X QH dose or 615 mg/kg). The concurrent control group received vehicle solution. The quantities of food and water consumed by each group of mice were monitored daily, and the body weights of the mice were measured weekly.

### 2.2. QH Supplementation

The daily-recommended intake dose of QH for humans is 500 mg/day. To convert the human dose to the animal dose in mice, we assumed a human weight of 60 kg and a body surface area correction factor/conversion coefficient of 12.3. For the 1X QH dose, the mouse dose used was 500 (mg)/60 (kg) = 8.33, 8.33 × 12.3 (the conversion coefficient) = 102.46 mg/kg mouse dose, and a conversion coefficient of 12.3 was used to account for the difference in body surface area between a mouse and a human.

### 2.3. Forelimb Grip Strength

A grip strength meter (Model-RX-5; Aikoh Engineering, Nagoya, Japan) was used to measure the forelimb grip strength. As a mouse grasps the metal bar (2 mm in diameter, 7.5 cm long), the tensile force is recorded on a digital force transducer. The detailed procedure was reported in our previous publication [17]. The forelimb grip strength test was performed after treatment of the indicated QH supplementation for 4 weeks. The test was performed 10 consecutive times, and the highest value during each trial was recorded with the attached force gauge. The mean measure of maximal force (in grams) recorded using this low-force system was used as the forelimb grip strength.

### 2.4. Exhaustive Swimming Exercise

Endurance performance is an important parameter for evaluating anti-fatigue, evaluated by an exhaustive swimming test. On the 29th day of the experiment, 30 min after the oral administration, the mice were placed individually in a columnar swimming pool (65 cm high with a 20 cm radius) containing fresh water maintained at 28 ± 1 °C, approximately 40 cm deep. Each mouse was loaded with a lead block, weighting approximately 5% of the body weight tagged to the tail. The endurance performance of each mouse was measured as the swimming time, recorded from the beginning to exhaustion. The mice were determined to be exhausted when uncoordinated movements occurred with failure to swim to the surface within a 7-second period [18]. The exhaustive swimming time was used as an index of exercise endurance.

### 2.5. Blood Biochemical Indices Upon Acute 10-min Free Swimming Test

The effects of QH on serum lactate levels were evaluated after 4 weeks of administration. The fatigue-related variables were assessed under fasting conditions to reflect the real physiological adaptation during the acute exercise challenge. The blood sample time points were pre-exercise, after 10 min of the swimming exercise, and following 20 min of rest. Serum samples were collected after centrifugation at 1000× *g* for 15 min at 4 °C and were analyzed by an autoanalyzer (Hitachi 7060, Hitachi, Tokyo, Japan).

### 2.6. The 90-min Free Swimming Test with Serum Biochemical Measurements

On the 33rd day of the experiment, 30 min after the oral administration, mice from each group were subjected to a 90-min free swimming challenge without a weight load. At the end of the swim, the mice rested for 60 min before blood samples were taken for analysis of urea nitrogen (BUN), creatine kinase (CK), ammonia (NH_3_) and free fatty acids (FFA). Serum samples were collected after centrifugation at 1000× *g* for 15 min at 4 °C and were analyzed by an autoanalyzer (Hitachi 7060, Hitachi, Tokyo, Japan).

### 2.7. Tissue Sample Preparation

All animals were euthanized with 95% CO_2_ asphyxiation 48 h after the last treatment, and blood was immediately collected. After the mice were euthanized, the liver, kidneys, heart, lung, skeletal muscle, epididymal fat pad (EFP), and brown adipose tissue (BAT) were accurately excised and weighed. After skin removal, the hind limb muscle was separated from the tibia and fibula bones. The gastrocnemius muscle was isolated from its surrounding muscles. Organs and tissues were carefully excised, rinsed in saline solution and blotted dry. The whole weight and the specific tissue weight (%) relative to the individual body weight were recorded and calculated. The muscle and liver tissues were collected immediately after saline cleaning and stored in liquid nitrogen for glycogen concentration analysis. 

### 2.8. Blood Biochemical Assessments

After the experiments, all mice were euthanized by 95% CO_2_ asphyxiation one hour after the last treatment, and blood was obtained by cardiac puncture blood centrifuged at 1000× *g* for 15 min at 4 °C for serum preparation. For a clinical biochemical assessment, aspartate aminotransferase (AST), alanine aminotransferase (ALT), albumin, total protein (TP), creatinine, blood urea nitrogen (BUN), creatinine, creatine kinase (CK), lactate dehydrogenase (LDH), UA, TG, TC and glucose levels were analyzed in an autoanalyzer (Hitachi7060, Hitachi, Tokyo, Japan).

### 2.9. Glycogen Concentration Analysis

Parts of the liver and muscle tissues were stored in liquid nitrogen for glycogen concentration analysis, as described previously [19]. Briefly, 100 mg of liver and muscle tissue was finely cut, weighed, and homogenized in 0.5 cold perchloric acid. After centrifugation for 15 min at 15,000× *g* and 4 °C, the supernatant was discarded. Standard glycogen (Sigma, USA) or tissue extracts (30 µL) were added to wells of a 96-well plate, followed by an iodine-potassium iodide reagent (200 µL). The plate was allowed to rest for 10 min before the absorbance was measured at 460 nm by an ELISA reader. 

### 2.10. Histological Staining of Tissues

Organs were carefully removed, minced, and fixed in 10% formalin. Tissues were embedded in paraffin and cut into 5 μm thick slices for morphological and pathological evaluation. Sections were stained with hematoxylin-eosin (H&E) and examined under a microscope equipped with a charged couple device (CCD) camera (BX-51, Olympus, Tokyo, Japan). 

### 2.11. Periodic Acid–Schiff Staining

Intracellular liver and muscle glycogen were stained with the periodic acid Schiff (PAS) staining solution (Muto Pure Chemicals, Tokyo, Japan) according to the standard protocol. Briefly, sections were deparaffinized and reacted with 1% periodic acid solution. The samples were then reacted with Schiff’s reagent and washed using NaHSO_3_/HCl solution. Hematoxylin was used for counterstaining. The samples were examined under a light microscope equipped with a CCD camera (BX-51, Olympus, Tokyo, Japan).

### 2.12. Statistical Analysis

Data are expressed as the mean ± SEM. All the statistical analyses were performed using SAS 9.4 (SAS Inst., Cary, NC, USA) based on one-way analysis of variance (ANOVA). The Cochran–Armitage test was used for the dose-effect trend analysis. The level of statistical significance was set at *p* < 0.05.

## 3. Results

### 3.1. Effects of QH Supplementation on Body Weight, Food and Water Consumption, and Organ Weight

The body weights, food consumption, and body compositions are summarized in Table 1 and Figure 2. There was no significant difference in body weight, food intake and water intake among the vehicle, QH-1X, QH-2X, and QH-6X groups (Figure 2). 

The muscle weights of the QH-1X, QH-2X, and QH-6X groups were significantly higher than those of the vehicle group, by 1.08- (*p* = 0.0089), 1.09- (*p* = 0.0044) and 1.07-fold (*p* < 0.0176), respectively. The BAT weights of the QH-1X, QH-2X, and QH-6X groups were significantly higher than those of the vehicle group, by 1.32- (*p* = 0.0008), 1.36- (*p* = 0.0003), and 1.41-fold (*p* < 0.0001), respectively. In the trend analysis, the muscle and BAT weight dose-dependently increased as the QH dose (*p* = 0.0156, *p* < 0.0001) increased.

The relative muscle weights (%) were higher in the QH-1X, QH-2X, and QH-6X groups than in the vehicle group, by 1.10- (*p* = 0.0065), 1.11- (*p* = 0.0032), and 1.11-fold (*p* = 0.0052), respectively. The relative BAT weight (%) was higher in the QH-1X, QH-2X, and QH-6X groups than in the vehicle group, by 1.35- (*p* = 0.0012), 1.40- (*p* = 0.0003) and 1.45-fold (*p* < 0.0001), respectively. In the trend analysis, the relative muscle and BAT weight (%) dose-dependently increased as the QH dose (*p* = 0.0006, *p* < 0.0001) increased.

### 3.2. Effects of QH on Grip Strength

In Figure 3A, the forelimb grip strengths of the vehicle, QH-1X, QH-2X, and QH-6X groups were 140 g, 172 g, 180 g, and 185 g, respectively. These results represent increases in grip strength of 1.23-, 1.29-, and 1.33-fold (all *p* < 0.0001) in the QH-1X, QH-2X, and QH-6X groups, respectively, compared with the vehicle group. The effect of QH on forelimb grip strength was shown to be dose-dependent (*p* < 0.0001). The relative grip strengths of the vehicle (363 ± 15%), QH-1X (457 ± 7%), QH-2X (476 ± 10%), and QH-6X (493 ± 11%) groups significantly increased by 1.26-, 1.31-, and 1.36-fold (all *p* < 0.0001), respectively, compared with the vehicle group. Relative grip strength (%), calculated by normalization to individual body weight, was higher in the QH treatment groups (Figure 3B) than in the vehicle group, and the trend was significant (*p* < 0.0001).

### 3.3. Effects of QH Supplementation on Endurance Capacity in the Exhaustive Swimming Test

In the swimming test, the exercise endurance levels of the vehicle, QH-1X, QH-2X, and QH-6X mice were 5.37, 11.47, 14.67, and 16.85 min, respectively, as shown in Figure 4. QH supplementation enhanced the swimming capability by delaying the onset of physical fatigue in the mice by 2.14- (*p* = 0.0128), 2.73- (*p* = 0.0003), and 3.14-fold (*p* < 0.0001), with increases in the QH dose from 1X and 2X to 6X, respectively. In addition, there was a significant dose-dependent effect on the maximal swim time (*p* < 0.0001).

### 3.4. Effects of QH Supplementation on Lactate after a 10-min Swimming Test

Blood lactate metabolite levels remain high during exercise physiological status. Lactate levels were assessed at three time points: pre-exercise, immediately post-exercise, and after rest (three time points) within the QH treatments (Table 2). Before swimming, there were no significant differences in the levels of blood lactate among the vehicle, QH-1X, QH-2X, and QH-6X groups. After 10 min of swimming, the levels of blood lactate were significantly lower in the QH-1X, QH-2X, and QH-6X groups than in the vehicle group, by 22.2%, 26.2%, and 27.9% (all *p* < 0.0001), respectively. After 20 min of recovery, the levels of blood lactate were significantly lower in the QH-1X, QH-2X, and QH-6X groups than in the vehicle group, by 24.5%, 24.8%, and 27.9% (all *p* < 0.0001), respectively. The effects of QH on lactate levels post-exercise and after 20 min rest were dose-dependent (*p* < 0.0001).

The lactate ratios before (pre) and after (post) the swimming exercise demonstrated that the accumulation of lactate was significantly lower in the QH-1X, QH-2X and QH-6X groups than in the vehicle group, by 37.6%, 42.0%, and 43.8% (all *p* < 0.0001), respectively. The rates of lactate clearance were 1.60- (*p* = 0.0107), 1.65- (*p* = 0.0066), and 1.67-fold (*p* = 0.0049) higher in the respective QH-1X, QH-2X, and QH-6X groups than in the vehicle group. Overall, QH treatment suppressed lactate production immediately after the swimming exercise and improved the clearance of lactate during the recovery period. 

### 3.5. Effects of QH Supplementation on BUN, CK, Ammonia, and FFA after a 90-min Swimming Test and a 60-min Rest Period

The BUN, CK, ammonia, and FFA levels in the blood of the mice groups were measured. The measurements were conducted after 90 min of swimming and 60 min of rest with a constant workload, and the results are shown in Figure 5. Serum BUN concentrations were significantly lower in the QH-1X, QH-2X, and QH-6X groups than in the vehicle group, by 8.7% (*p* = 0.0293), 11.4% (*p* = 0.0053), and 24.5% (*p* = 0.0053), respectively (Figure 5A). As shown in Figure 5B, the serum CK levels were significantly lower in the QH-1X, QH-2X, and QH-6X groups than in the vehicle group, by 8.7% (*p* = 0.0293), 11.4% (*p* = 0.0053), and 24.5% (*p* < 0.0001), respectively. The levels of ammonia were significantly lower in the QH-1X, QH-2X, and QH-6X groups than in the vehicle group, by 28.7% (*p* = 0.0004), 32.1% (*p* < 0.0001), and 33.0% (*p* < 0.0001), respectively (Figure 5C). Figure 5D shows that levels of serum FFA were significantly higher in the QH-1X, QH-2X, and QH-6X groups than in the vehicle group, by 1.16- (*p* = 0.0127), 1.28- (*p* < 0.0001), and 1.32-fold (*p* < 0.0001), respectively. The trend analysis of QH supplementation indicated dose-dependent decreases in the serum levels of BUN, CK, and ammonia and increases in the level of FFA (all *p* < 0.0001).

### 3.6. Effects of QH Supplementation on Biochemical Assessments

Biochemical results at the end of the experiment provided clinical information about the health status of the test animals. The levels of biochemical indices, including AST, ALT, albumin, TP, BUN, CK, TC, and glucose, did not differ among groups (*p* > 0.05, Table 3). The BUN levels were significantly lower in the QH-1X, QH-2X, and QH-6X groups than in the vehicle group, by 14.4% (*p* = 0.0085), 17.4% (*p* = 0.0018), and 18.61% (*p* = 0.0010), respectively. The levels of creatinine were significantly lower in the QH-2X and QH-6X groups than in the vehicle group, by 18.7% (*p* = 0.0213) and 18.3% (*p* = 0.0234), respectively. Serum UA concentrations were significantly lower in the QH-1X, QH-2X, and QH-6X groups than in the vehicle group, by 25.0% (*p* = 0.0048), 26.8% (*p* = 0.0027), and 29.2% (*p* = 0.0012), respectively. Compared with that in the vehicle group, serum LDH levels in the QH-1X, QH-2X, and QH-6X groups decreased by 17.3% (*p* = 0.0182), 17.4% (*p* = 0.0173), and 22.1% (*p* = 0.0032), respectively. The serum levels of TG were significantly lower in the QH-2X and QH-6X groups than in the vehicle group, by 22.4% (*p* = 0.0010) and 23.47% (*p* = 0.0006), respectively. Trend analysis indicated that serum BUN (*p* = 0.0007), creatinine (*p* = 0.0155), UA (*p* = 0.0017), LDH (*p* = 0.0024), and TG (*p* < 0.0001) levels were dose-dependently decreased by QH supplementation.

### 3.7. Effects of QH Supplementation on Tissue Glycogen Determination

The glycogen concentration in the livers and skeletal muscles of the mice are shown in Figure 6. As shown in Figure 6A, the QH-1X, QH-2X, and QH-6X groups showed significantly elevated glycogen stores in the liver, which were increased by 2.25- (*p* = 0.0006), 2.63- (*p* < 0.0001), and 3.14-fold (*p* < 0.0001), respectively, as compared with the vehicle group. Figure 6B shows that the muscle glycogen levels of the QH-6X group were 1.66- (*p* = 0.0013), 1.62- (*p* = 0.0017), and 1.49-fold (*p* = 0.0062) higher than those of the respective vehicle, QH-1X, and QH-2X groups. There was no significant difference between the vehicle, QH-1X, and QH-2X groups. The trend analysis revealed that glycogen stores in the liver and muscle (all *p* < 0.0001) underwent a significant dose-dependent effect.

### 3.8. Effects of QH Supplementation on Histology and PAS stain

As seen in Figure 7, the four groups did not differ in the histological observations of the liver, kidney, heart, lung, muscle, and EFP. The arrangement of sinusoid and hepatic cords in the liver showed no changes with respect to QH treatments (Figure 7A). Hyperplasia and hypertrophy were not observed in rhabdomyocytes of the gastrocnemius muscle (Figure 7B) or heart cardiomyocytes (Figure 7C). All animals showed typical tissue architectures of the lung alveoli by H&E staining (Figure 8D). In addition, adipose tissue morphology and fat cell size did not differ between the groups (Figure 7E). The structures of renal tubules and the glomerulus did not differ between the groups (Figure 7F). The results suggest that QH supplementation at these doses was safe and there were no apparent damage effects on major organs or tissues. 

PAS staining can be used to provide a visual and qualitative assessment of the glycogen concentration, wherein more intensely pink-stained fibers and muscles contain more glycogen (Figure 8). Compared with those of the vehicle group, the liver and muscle tissues of the mice in the QH-2X and QH-6X groups demonstrated higher amounts of glycogen. Overall, PAS staining confirmed the increases in the liver and muscle glycogen resulting from QH treatments.

## 4. Discussion

Ubiquinol (QH) plays a major role in cell energy production and has both energy production and energy activating effects [7]. In our previous study, muscle strength was positively correlated with grip strength [17]. In this study, we observed greater grip strength in the QH-6X group than in the other groups. A previous study reported that CoQ10 supplementation resulted in increased short-term maximum performance, perhaps via an increase in ATP and creatinine phosphate synthesis [20]. Our experimental results indicate that QH supplementation with strength training could increase muscle mass (Table 1) and benefit grip strength capacity (Figure 3).

Ubiquinol acts as a potent lipophilic antioxidant to protect against radical-induced lipid, protein, and DNA oxidation, which, in turn, promotes its use as a supplement to improve exercise performance, reduce exercise-induced muscular injury, and enhance plasma and cellular antioxidant levels [21,22]. The swimming test to exhaustion is an experimental exercise model for evaluating physical fatigue. In our study, the anti-fatigue activity of QH was demonstrated by the fact that the endurance time to exhaustion of the mice was extended significantly in the QH-supplemented groups (Figure 4). A previous study showed that suppression of the level of NFκB due to QH administration could decrease oxidative stress because QH acts as an antioxidant and free-radical scavenger [23]. Another study has suggested that chronic CoQ10 supplementation increases plasma CoQ10 concentrations and increases the time to exhaustion; the authors suggested that increases in exercise performance could be due to a combination of enhanced oxidative phosphorylation within the mitochondria and enhanced antioxidant protection [24]. Additionally, QH might have an anti-fatigue effect and could improve physical exercise capacity.

Several biochemical parameters have been used to evaluate the extent of exercise-induced fatigue and injury after exercise, some examples being lactate, BUN, CK, ammonia, and FFA [25]. Blood lactate is derived from the anaerobic metabolism of glucose during exercise [26]. This reduction of pH in the blood and muscle tissues leads to inhibition of muscle contraction and glycolysis, as well as various biochemical, metabolic, and physiological side effects [27]. In our study, the blood lactate levels after exercise were significantly lower in the QH-supplemented groups (Table 2). In addition, the rates of blood lactate clearance were also superior to those in the vehicle group. These results may indicate that QH increases the contribution of aerobic metabolism during exercise. Previous research has shown that long-term CoQ10 supplementation could significantly increase the plasma levels of CoQ10 oxidative phosphorylation after exhaustive exercise. The authors suggested that CoQ10 supplementation accelerates electron transfer, increases ATP production (adenosine triphosphate production) and decreases the accumulation of blood lactate [28,29]. Therefore, we suggest that 4 weeks of QH supplementation is beneficial to decrease exercise-induced lactate levels and production rate.

BUN is an important metabolite formed by protein degradation after intensive exercise. Urea is the main end product of protein catabolism and protein metabolism, and depending on the type of the exercise, is an important index correlated with dehydration, protein breakdown, fatigue, and stress [30]. In our study, we demonstrated that the BUN levels after exercise were significantly lower in the QH-supplemented groups. There is one report showing that decreased BUN levels, which reflect reduced protein metabolism and increased hepatic glycogen storage, may provide an extra energy source for mice during CoQ10 supplementation that results in improved physical stamina [31]. This agrees with our results; we found that long-term QH supplementation increased hepatic glycogen storage (Figure 6) and FFA levels after exercise (Figure 5D) and could provide an energy source during exercise to improve exercise performance.

CK activity in blood is an indirect marker of muscle damage, and it has been considered to be a useful indicator in sports medicine for the detection of muscle damage [32]. Marked reductions in CK levels after exercise were observed in the QH-supplemented groups in our study. Another study also demonstrated that CoQ10 supplementation reduced the adverse effects of CK enzyme activity on athletes after a session of exhausting aerobic exercise on a treadmill [32].

Ammonia is produced by skeletal muscle during short intense and prolonged submaximal exercise. Under these conditions, the rate of ATP utilization may exceed the rate of ATP production. To stabilize the relative ratios of ATP to ADP and AMP, AMP deaminase functions to remove excessive AMP. The breakdown of AMP to IMP in the AMP deaminase reaction produces ammonia [33]. Our study showed significantly lower ammonia levels after exercise in the QH-supplemented groups. Since QH participates in the production of ATP and can increase ATP levels in mitochondria, the activity of AMP deaminase may be suppressed and the production of ammonia thereby decreased.

Exercise increases the use of triacylglycerol, which is hydrolyzed to free fatty acids, which, in turn, are then released into the circulation and provide fuel for the working muscles [34]. In our study, we found significant increases in FFA levels after exercise in the QH-supplemented groups. CoQ10 is believed to decrease oxidative stress and participate in endothelial metabolism, resulting in an increase in lipolysis of triglycerides [35] and the availability of FFA. Our experimental results indicate that long-term QH supplementation increased FFA (Figure 5D) levels after exercise, possibly due to increased lipolysis, which may also explain the findings of significantly lower serum TG levels in the QH-supplemented groups (Table 3).

In this study, we showed that the BUN, creatinine, UA, LDH, and TG levels were lower in the QH-6X group than in the other groups (Table 3). A recent study showed that CoQ10 can reduce BUN and creatinine levels to alleviate radiation-induced nephropathy in rats [36]. Another study demonstrated that short-term ingestion of coenzyme Q10 may have modulated the LDH and CPK levels of a trial group of athletes, whose levels were observed to increase less than those of a control group of athletes [37]. A previous study reported that in ApoE^−/−^ HD mice, oral CoQ10 was likely responsible for reducing metabolic parameters, such as TC, low-density lipoprotein-cholesterol, and TG levels [38]. Additionally, ubiquinone supplementation has the potential to decrease severe kidney and muscle injuries and fat metabolism.

Liver and skeletal muscle glycogen are important sources of energy storage and supply the key metabolic regulators of exercise [39]. Increases in glycogen storage in the liver and muscles can enhance physical endurance [17]. In our study, we found a dose-dependent increase in the glycogen concentration in the liver and muscles in the QH-supplemented groups. The results of the histological observations were in accordance with the results of the liver glycogen concentration, indicating that QH supplementation can increase the liver glycogen concentration (Figure 8). Wek et al. found that yeast strains containing respiration-deficient mutations in genes, such as COQ3, which is required for the synthesis of CoQ10, were reduced in their ability to accumulate glycogen due to inactivation of glycogen synthase [40]. This suggests that QH is essential in the regulation of glycogen synthase activity and glycogen storage. 

Glycogen, glucose, and free fatty acids provide the vast majority of fuel required for energy production in skeletal muscle during aerobic exercise [41]. Previous studies have pointed out that oxidation of FFA spares muscle glycogen and thus increases work endurance [42]. In addition, QH supplementation is beneficial for the promotion of the oxidation of TG to FFA during exercise. Another study also demonstrated an increase in fat oxidation through a rise in AMPK activation, leading to spare glycogen storage [43]. Our results suggest that QH treatment is an effective method for the oxidation of TG, increasing the FFA concentration while decreasing hepatic and muscle glycogen utilization to enhance exercise performance capacity. We suggest that QH treatment could be a new sport nutrition supplementation for the prevention of fatigue. Further studies are needed to completely understand the function and to explore the exact mechanisms of the anti-fatigue effects of QH on humans.

## 5. Conclusions

The results from this study suggest that 4 weeks of QH supplementation can significantly improve physical activities, including grip strength and endurance capacity, by reducing metabolites, including lactate, BUN, CK, and ammonia, and by improving the muscle and liver glycogen synthesis and reserves. This effect appears to be related to a switch to lipid utilization as an energy source. Therefore, QH is an emerging molecule in sport nutrition supplementation to elevate exercise performance and tolerance and to reduce physical fatigue in exhaustive exercise and under modern life stress conditions. In conclusion, these results suggest that QH formulation should not lead to any safety concerns and can mitigate fatigue and enhance exercise performance.

## Figures and Tables

**Figure 1 nutrients-11-02550-f001:**
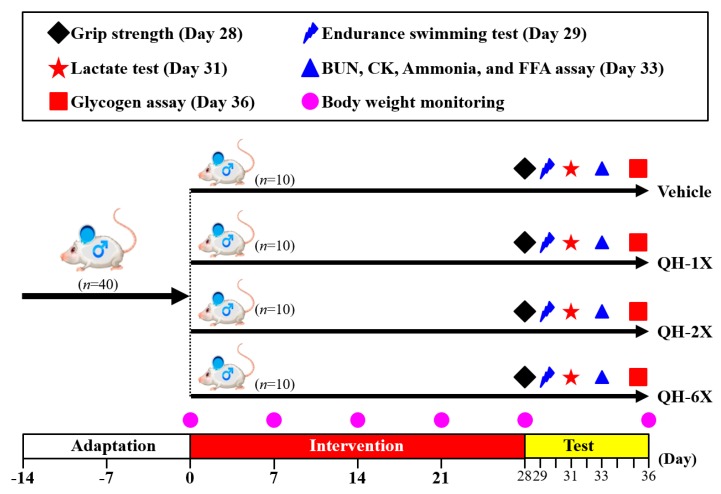
Experimental designs for the effects of ubiquinol (QH) on exercise adaptation. The animals were randomly assigned to the four groups indicated (vehicle, QH-1X, QH-2X, and QH-6X) and were consecutively supplemented with QH until the end of the experiments. The physical capacities and related biochemistries were assessed within the test duration.

**Figure 2 nutrients-11-02550-f002:**
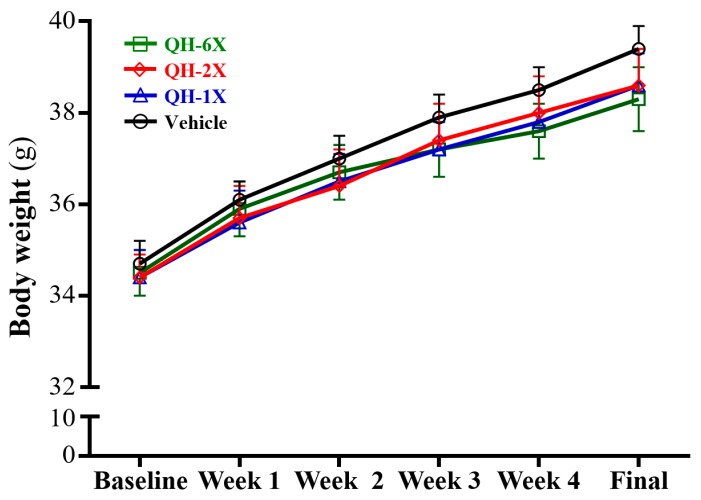
The effect of QH supplementation on growth curves. Data are the mean ± SEM for *n* = 10 mice in each group.

**Figure 3 nutrients-11-02550-f003:**
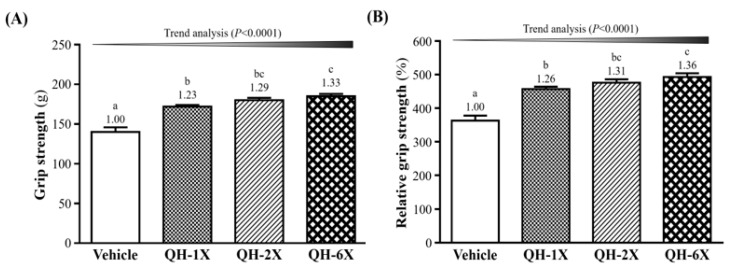
Effect of 4-week QH supplementation on (**A**) forelimb grip strength; (**B**) relative grip strength (%). Data are the mean ± SEM for *n* = 10 mice per group. Different letters (a, b, c) indicate significant differences at *p* < 0.05 according to one-way ANOVA.

**Figure 4 nutrients-11-02550-f004:**
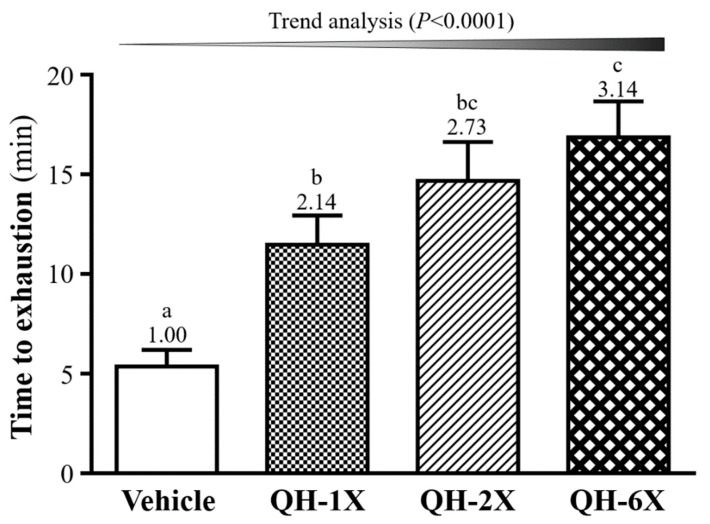
Effect of 4 weeks of QH supplementation on exhaustive swimming test. Data are the mean ± SEM for *n* = 10 mice in each group. Different letters (a, b, c) indicate significant differences at *p* < 0.05.

**Figure 5 nutrients-11-02550-f005:**
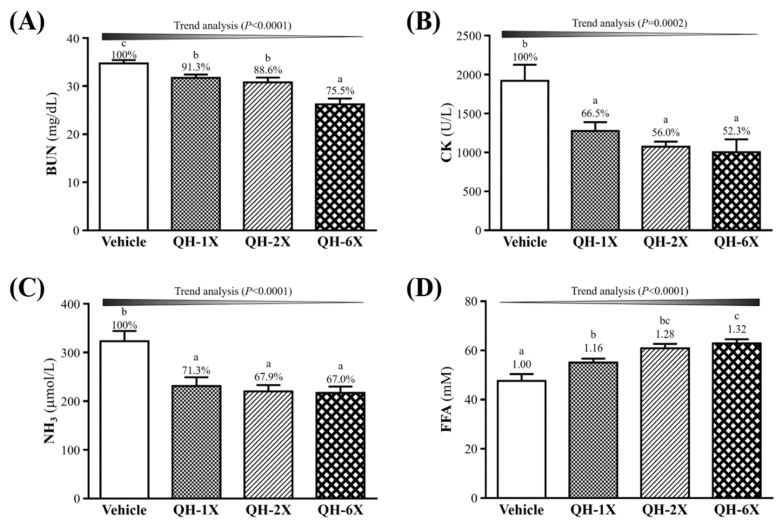
Effects of QH supplementation on serum (**A**) blood urea nitrogen (BUN); (**B**) creatine kinase (CK); (**C**) ammonia; and (**D**) free fatty acid (FFA) levels after an extended exercise challenge. The four groups indicated underwent 90 min of swimming exercise, and blood was sampled after 60 min of rest. Data are the mean ± SEM for *n* = 10 mice in each group. Different letters indicate significant differences at *p* < 0.05 according to one-way ANOVA.

**Figure 6 nutrients-11-02550-f006:**
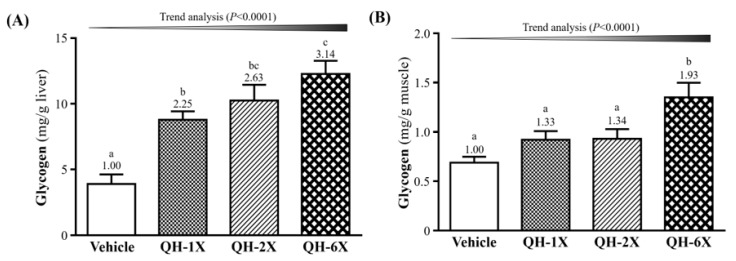
Effects of QH supplementation on the serum glycogen concentration in (**A**) hepatic and (**B**) muscle. Data are the mean ± SEM for *n* = 10 mice per group. Different letters indicate significant differences at *p* < 0.05.

**Figure 7 nutrients-11-02550-f007:**
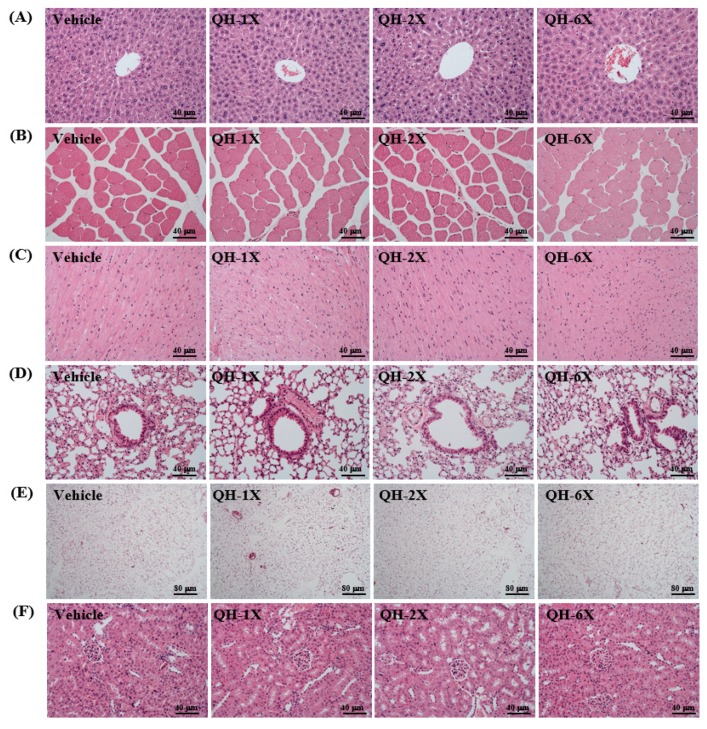
Effects of QH supplementation on the histomorphology of (**A**) liver, (**B**) skeletal muscle, (**C**) heart, (**D**) lung, (**E**) epididymal fat pad, and (**F**) kidney in mice. Specimens were photographed using light microscopy. Hematoxylin-eosin (H&E) stain, magnification: 100× or 200×; scale bar, 40 or 80 μm.

**Figure 8 nutrients-11-02550-f008:**
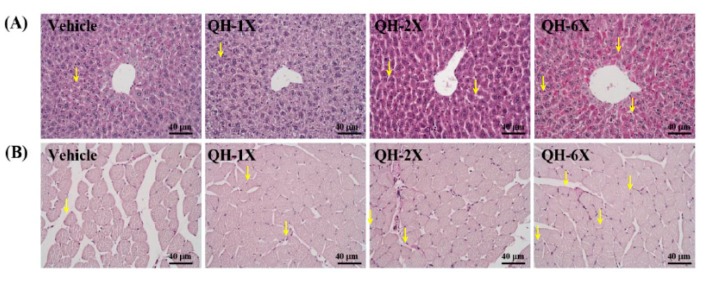
Effects of QH supplementation on PAS staining of (**A**) liver and (**B**) skeletal muscle. Specimens were photographed using light microscopy. H&E stain, magnification: 200x; scale bar, 40 μm. Glycogen indicated by yellow arrows (↓).

**Table 1 nutrients-11-02550-t001:** General characteristics of mice with QH supplementation.

Characteristic	Vehicle	QH-1X	QH-2X	QH-6X	Trend Analysis
Initial BW (g)	34.7 ± 0.5	34.4 ± 0.6	34.4 ± 0.5	34.5 ± 0.5	0.7737
Final BW (g)	39.4 ± 0.5	38.6 ± 0.7	38.6 ± 0.8	38.3 ± 0.7	0.1351
Food intake (g/day)	6.5 ± 0.3	6.8 ± 0.4	6.6 ± 0.3	6.8 ± 0.3	0.6415
Water intake (mL/day)	7.7 ± 0.3	7.5 ± 0.3	7.8 ± 0.3	7.8 ± 0.2	0.541
Liver (g)	2.22 ± 0.05	2.20 ± 0.08	2.20 ± 0.07	2.13 ± 0.05	0.227
Muscle (g)	0.37 ± 0.007 ^a^	0.40 ± 0.006 ^b^	0.40 ± 0.010 ^b^	0.40 ± 0.008 ^b^	0.0156
Kidney (g)	0.58 ± 0.030	0.55 ± 0.025	0.56 ± 0.016	0.57 ± 0.032	0.4179
Heart (g)	0.24 ± 0.006	0.24 ± 0.008	0.22 ± 0.010	0.22 ± 0.011	0.1229
Lung (g)	0.23 ± 0.006	0.23 ± 0.004	0.22 ± 0.003	0.23 ± 0.004	0.785
EFP (g)	0.48 ± 0.021	0.50 ± 0.029	0.48 ± 0.004	0.40 ± 0.036	0.2067
BAT (g)	0.09 ± 0.003 ^a^	0.12 ± 0.005 ^b^	0.12 ± 0.007 ^b^	0.12 ± 0.006 ^b^	<0.0001
Relative liver weight (%)	5.65 ± 0.14	5.69 ± 0.17	5.71 ± 0.15	5.59 ± 0.14	0.4766
Relative muscle weight (%)	0.94 ± 0.015 ^a^	1.04 ± 0.018 ^b^	1.05 ± 0.026 ^b^	1.04 ± 0.026 ^b^	0.0006
Relative kidney weight (%)	1.47 ± 0.07	1.43 ± 0.06	1.46 ± 0.04	1.48 ± 0.07	0.517
Relative heart weight (%)	0.61 ± 0.014	0.63 ± 0.021	0.57 ± 0.023	0.58 ± 0.020	0.1545
Relative lung weight (%)	0.57 ± 0.013	0.59 ± 0.014	0.57 ± 0.014	0.59 ± 0.014	0.4814
Relative EFP weight (%)	1.21 ± 0.05	1.29 ± 0.06	1.25 ± 0.10	1.05 ± 0.09	0.2263
Relative BAT weight (%)	0.22 ± 0.008 ^a^	0.30 ± 0.012 ^b^	0.31 ± 0.021 ^b^	0.32 ± 0.014 ^b^	<0.0001

Data are expressed as the mean ± SEM (*n* = 10). Values in the same row with different superscript letters (a, b) differed significantly, *p* < 0.05, according to one-way ANOVA. EFP: epididymal fat pad; BAT: brown adipose tissue.

**Table 2 nutrients-11-02550-t002:** The effects of QH on blood lactate level changes during the acute exercise challenge.

Time Point	Vehicle	QH-1X	QH-2X	QH-6X	Trend Analysis
	Lactate (mmol/L)				
Before swimming (A)	2.5 ± 0.1	2.5 ± 0.1	2.5 ± 0.1	2.5 ± 0.1	0.8238
After swimming (B)	6.8 ± 0.3 ^b^	5.3 ± 0.2 ^a^	5.0 ± 0.2 ^a^	4.9 ± 0.1 ^a^	<0.0001
After a 20-min rest (C)	5.2 ± 0.4 ^b^	3.3 ± 0.2 ^a^	3.0 ± 0.2 ^a^	2.9 ± 0.2 ^a^	<0.0001
	Rate of lactate production and clearance				
Production rate = B/A	2.75 ± 0.14 ^b^	2.07 ± 0.08 ^a^	2.06 ± 0.09 ^a^	1.98 ± 0.07 ^a^	<0.0001
Clearance rate = (B-C)/B	0.23 ± 0.03 ^a^	0.38 ± 0.03 ^b^	0.39 ± 0.04 ^b^	0.39 ± 0.04 ^b^	0.0076

The metabolite lactate was assessed for the four groups (vehicle, QH-1X, QH-2X, and QH-6X) at three points. The lactate production rate (B/A) was calculated as the lactate level after exercise (B) divided by the lactate level before exercise (A). The lactate difference between after exercise and after rest divided by after rest was defined as the clearance rate. Values in the same row with different superscript letters (a, b) differed significantly, *p* < 0.05.

**Table 3 nutrients-11-02550-t003:** Effect of 4-week QH supplementation on biochemical indices at the end of the study.

Characteristics	Vehicle	QH-1X	QH-2X	QH-6X	Trend Analysis
AST (U/L)	94 ± 7	90 ± 4	86 ± 5	85 ± 4	0.2805
ALT (U/L)	54 ± 3	50 ± 4	47 ± 4	45 ± 5	0.0125
Albumin (g/dL)	3.2 ± 0.1	3.3 ± 0.1	3.3 ± 0.1	3.3 ± 0.1	0.4777
TP (g/dL)	5.3 ± 0.0	5.3 ± 0.0	5.3 ± 0.1	5.3 ± 0.1	0.9616
BUN (mg/dL)	22.7 ± 1.1 ^b^	19.4 ± 0.9 ^a^	18.7 ± 0.5 ^a^	18.5 ± 0.8 ^a^	0.0007
Creatinine (mg/dL)	0.31 ± 0.02 ^b^	0.27 ± 0.02 ^ab^	0.25 ± 0.02 ^a^	0.25 ± 0.01 ^a^	0.0155
UA (mg/dL)	0.96 ± 0.05 ^b^	0.72 ± 0.06 ^a^	0.70 ± 0.04 ^a^	0.68 ± 0.07 ^a^	0.0017
CK (U/L)	327 ± 48	315 ± 39	328 ± 25	322 ± 34	0.8329
LDH (U/L)	534 ± 28 ^b^	442 ± 23 ^a^	441 ± 31 ^a^	416 ± 23 ^a^	0.0024
TC (mg/dL)	150 ± 5	149 ± 4	146 ± 4	139 ± 4	0.0553
TG (mg/dL)	162 ± 8 ^b^	158 ± 6 ^b^	126 ± 8 ^a^	124 ± 7 ^a^	<0.0001
Glucose (mg/dL)	115 ± 3	113 ± 3	114 ± 4	114 ± 3	0.5234

Data are expressed as the mean ± SEM (*n* = 10). Values in the same row with different superscript letters (a, b) differed significantly, *p* < 0.05. ALT, alanine aminotransferase; AST, aspartate aminotransferase; TP, total protein; BUN, blood urea nitrogen; CK, creatine kinase; UA, uric acid; LDH, lactate dehydrogenase; TC, total cholesterol; TG, triacylglycerol.

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
