# Peer review of "Ubiquinol Supplementation Alters Exercise Induced Fatigue by Increasing Lipid Utilization in Mice"

_nutrients, 2019, doi:10.3390/nu11112550_

Round 1

Reviewer 1 Report

In this work, Huan-Chieh and colleagues assessed the effects of 1 month administration of ubiquinol (QH), the reduced -and more bioavailable-form of CoQ on physical fatigue, in vivo. The study included appropriate number of animals, assessment of different doses, and biochemical and functional outcome measures.

Although the study might have clinical relevance, it is not particularly novel, since the authors mention that antioxidants, including CoQ, have been reported to be effective in physical fatigue.

Furthermore, the study lacks experiments aimed to understand whether QH supplementation indeed reduces oxidative stress and/or rescues ATP synthesis.

Since the main goal of the study is to prove that QH is a better alternative to CoQ based on its bioavailability, the authors should measure QH in plasma and organ intake, or at least muscle, before and after administration of the three doses.

Specific comments

-Page 1 lines 60-61, the sentence “CoQ10 is a large molecule” is repeated twice

-Page 1 line 62: it is not clear why the authors quote ref 13 and what they mean by “ poor CoQ absorption with low plasma level”. The limitation of CoQ is its poor organ uptake; high doses of CoQ lead to accumulation in plasma and liver. The authors should quote a study that investigated CoQ administration in patients with muscle fatigue, rather than hearth failure.

-Page 2 line 98: “The administration dose of QH for human daily-recommended intake is 500 mg/Kg”. This needs reference/s. Also, for what condition?

-Page 3, line 137: ”killed” should be “euthanized”

-Page 4, line 171: I think SD is more appropriate than SEM for comparison of different animals

I don’t understand the representation of significant differences with letters. I would like to see the significant differences between treatments and placebo. I understand that the outcomes improve with increasing doses, but is the increment statistically relevat?

-Figure 6 is missing

-Page 11, line 350: The levels of CoQ in plasma do not reflect the levels in muscle. However, if the authors think that CoQ is effective, why try QH?

-Page 11, line: 353 “We suggest that long-term QH….and production rate”.  In this study, QH was administered for 1 month, therefore they can make assumptions only on short-term treatment.

Reviewer 2 Report

There are a number of suggested changes in comments in the paper. Clearly the major finding in this paper is the switch to higher lipid utilization with ubiquinol supplementation. The higher lipid utilization reduces glycogen and muscle protein utilization as energy sources. You need to discuss this in greater length in the discussion and possibly to a greater degree in the introduction. 

Some of the tables will need the data changed to increase the number of decimal points to expose the changes. e.g. table 1. all three UQ groups are 0.40+/-0.01 yet they all have different p-values. An increase in one decimal place may remove the readers confusion as they appear the same.

You need a study limitations paragraph at the end of the discussion.

I would suggest changing the title to reflect the type of fatigue being assessed (exercise induced muscle fatigue) and not simply the word fatigue. I would also consider putting the major finding in the title to attract more readers. example "Ubiquinol supplementation alters exercise induced fatigue by increasing lipid utilization in mice".

Round 2

Reviewer 1 Report

The authors partially addressed the issues raised by the reviewers. Overall the manuscript improved

Author Response

"Ubiquinol supplementation alters exercise induced fatigue by increasing lipid utilization in mice (nutrients-602741)"

Modifications and revisions

Thank you very much for the reviewer’s valuable comments. The paper was revised on the basis of the constructive suggestions.

Response to Reviewer’s Comments

For Reviewer:

The authors partially addressed the issues raised by the reviewers. Overall the manuscript improved.

Response: Thank you very much for the reviewer’s valuable comments. This article has been revised and edited by MDPI.
